# CoAnnotating: Uncertainty-Guided Work Allocation between Human and Large Language Models for Data Annotation

**Minzhi Li** [†§]     **Taiwei Shi** [‡]     **Caleb Ziems** [¶]

**Min-Yen Kan** [†]     **Nancy F. Chen** [§]     **Zhengyuan Liu** [§]     **Diyi Yang** [¶]

[†]National University of Singapore     [§]Institute for Infocomm Research (I²R), A*STAR
[‡]University of Southern California     [¶]Stanford University

li.minzhi@u.nus.edu     taiweish@usc.edu     cziems@stanford.edu

nfychen@i2r.a-star.edu.sg     liu_zhengyuan@i2r.a-star.edu.sg

kanmy@comp.nus.edu.sg     diyiy@cs.stanford.edu

## Abstract

Annotated data plays a critical role in Natural Language Processing (NLP) in training models and evaluating their performance. Given recent developments in Large Language Models (LLMs), models such as ChatGPT demonstrate zero-shot capability on many text-annotation tasks, comparable with or even exceeding human annotators. Such LLMs can serve as alternatives for manual annotation, due to lower costs and higher scalability. However, limited work has leveraged LLMs as complementary annotators, nor explored how annotation work is best allocated among humans and LLMs to achieve both quality and cost objectives. We propose *CoAnnotating*, a novel paradigm for Human-LLM co-annotation of unstructured texts at scale. Under this framework, we utilize uncertainty to estimate LLMs' annotation capability. Our empirical study shows *CoAnnotating* to be an effective means to allocate work from results on different datasets, with up to 21% performance improvement over random baseline. For code implementation, see https://github.com/SALT-NLP/CoAnnotating.

## 1 Introduction

Labeled data plays a critical role in establishing benchmarks and developing models for Natural Language Processing (NLP). Although Large Language Models (LLMs) like ChatGPT have demonstrated their strong zero-shot performance in various tasks such as question answering, reasoning, natural language inference, sentiment analysis, and named entity recognition, results obtained by fine-tuned language models still outperform LLMs on most of these tasks (Qin et al., 2023; Zhong et al., 2023; Ziems et al., 2023). Therefore, collecting labeled data for model training and fine-tuning is still valuable. Instead of deploying LLMs directly for downstream uses, it is worthwhile to investigate how researchers can leverage LLMs' zero-shot capability in labeling text data to construct high-

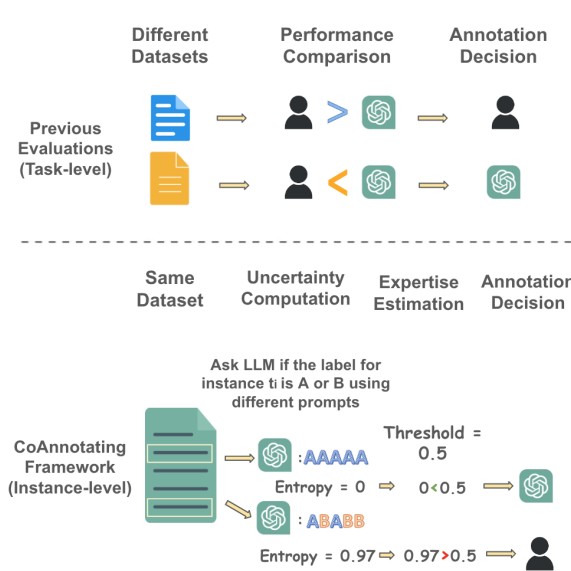

Figure 1: *CoAnnotating* framework. It differs from previous work by considering how to allocate data within the **same** dataset to humans and ChatGPT by obtaining responses from ChatGPT using different variations of prompts and estimating ChatGPT's annotation expertise with the use of uncertainty metrics such as entropy.

quality datasets and improve the performance of fine-tuned models.

Typically, researchers recruit human annotators such as experts or crowd workers to perform data annotation (Kittur et al., 2008; Snow et al., 2008). Some challenges in manual annotation includes high costs of recruiting and training annotators, annotation inconsistency and human subjectivity (Lingren et al., 2014; Grosman et al., 2020). Recent work explored how LLMs perform relative to crowd workers (Ding et al., 2022) and results showed that it is possible for LLMs like ChatGPT to replace large-scale manual annotation (Huang et al., 2023; Kuzman et al., 2023). In some cases, LLMs' annotation quality even outperforms human annotators on certain tasks (Gilardi et al., 2023). Given the much lower annotation cost than crowd workers, LLMs are considered to have great poten-

tial to increase the cost efficiency of the data annotation process. However, some studies also show that, relative to human performance, LLMs' zero-shot performance falls short on more difficult and pragmatic tasks (Wang et al., 2021; Kocoń et al., 2023). They suggest that practitioners should use caution when using LLMs to annotate data (Reiss, 2023; Huang et al., 2023). Such prior works view humans and LLMs as **competitors**, measuring the accuracy of LLM labels as a replacement for human annotation, rather than considering how humans and LLMs might **collaborate** in an efficient manner. It is Human-LLM collaboration that motivates this work. We propose the *CoAnnotating* framework, which aims to balance the complementary profiles of humans and LLMs in terms of their respective annotation quality and cost.

Our work tackles the problem of Human-LLM co-annotation from a *resource allocation* perspective. Following Gentile et al. (2022), Diao et al. (2023) and Wang et al. (2021), we consider model confidence as a reliable signal for the model's expected performance. As we consider allocating a given datapoint for an LLM to annotate, we can use the inverse of the model's uncertainty to estimate our confidence in that allocation. Under *CoAnnotating*, we quantify LLMs' annotating expertise on the **instance-level** (estimating how well LLMs can annotate the given data point) beyond **task-level** (evaluating how LLMs performs on overall for each dataset). As such, a more informed allocation decision can be made with this fine-grained and contextualized instance-level perspective, rather than broad and coarse dataset-level expertise.

We show that our proposed method using the uncertainty of responses can achieve a more efficient and more accurate work allocation than the random allocation baseline. Our results also show that confidence scores generated by LLMs are generally well-calibrated but not always reliable. It is possible to outsource some annotation work to achieve human-level performance for more straightforward tasks like topic understanding. On the other hand, a tradeoff between annotation quality and annotation cost is inevitable for more nuanced tasks. Our framework establishes a guide to effectively allocate AI and human efforts in collaborative annotation, and in doing so, it provides key insights into the capacities of LLMs, as well as the nature of the tasks and data that remain outside these capacities.

## 2 Related Work

### 2.1 Weak Supervision

In a traditional supervised learning setting, every training data point is labeled by human annotators. However, acquiring manually annotated labels for training data can be prohibitively costly and time-consuming. Weak supervision helps to address the challenge using partially and imperfectly labeled data for training (Zhang et al., 2022). Weak supervision techniques obtain these noisy labels by tapping into heuristics (Ratner et al., 2017; Meng et al., 2018; Awasthi et al., 2020), feature annotation (Mann and McCallum, 2010), external knowledge bases (Hoffmann et al., 2011; Min et al., 2013), pretrained models (Bach et al., 2019; Zhang et al., 2021) and third-party tools (Lison et al., 2020). Moreover, weak supervision can be combined with the active learning framework (Gonsior et al., 2020) to select the most informative data to be annotated by humans and utilize weak supervision to decide noisy labels. Given LLMs' stunning zero-shot capabilities, our work explores the possibility of using them as a more efficient labeling source, thus freeing up resources to be reinvested in the research pipeline.

### 2.2 LLMs for Annotation

Most prior works frame the decision for human or LLM annotation as one of competition rather than collaboration between these modes. These show that LLMs like GPT-3 `davinci-003` have strong zero-shot sentiment analysis performance (Ding et al., 2022). ChatGPT (`gpt-3.5-turbo`) performs surprisingly well on automatic genre detection in under-resourced languages like Slovenian (Kuzman et al., 2023). ChatGPT can even achieve high accuracy on some of the most nuanced tasks like implicit hate speech detection (Huang et al., 2023). Similarly, GPT-4 is able to annotate texts that require reasoning and contextual knowledge and provide explanations that could facilitate interpretive research (Törnberg, 2023). These results show the great potential of LLMs as data annotation tools with just simple prompt design and without much manual labeling efforts (Kuzman et al., 2023).

However, there is still room to close significant performance gaps between LLMs' performance and existing fine-tuned baselines on some challenging tasks. LLMs struggle with named entity recognition (Ding et al., 2022; Qin et al., 2023), relational reasoning (Bang et al., 2023), affective

tasks (Kocoń et al., 2023; Amin et al., 2023) and semantic similarity tasks (Kocmi and Federmann, 2023; Wang et al., 2023). Moreover, it does not outperform fine-tuned baselines for generation tasks like question answering and text summarization (Tan et al., 2023; Wang et al., 2023). These works all take the perspective that LLMs and humans are competitors, making task-level comparisons between LLMs and humans/fine-tuned models for each dataset. Our work views LLMs and humans as potential collaborators, with the possibility to work with each other to annotate the same dataset.

## 2.3 Human-Machine Collaboration for Dataset Creation

The quality of the dataset and the cost of creating a dataset are two important but sometimes conflicting objectives in dataset creation. Previous work suggests a human-AI collaborative framework that utilizes language models' generation capability and human revision and evaluation skills (Tekiroglu et al., 2020; Yuan et al., 2021; Bartolo et al., 2021; Liu et al., 2022) to create valuable datasets of high quality. For cost efficiency, some have proposed averaging or majority vote over human and machine outputs (Chaganty et al., 2018; Ziems et al., 2023) and some initial empirical explorations such as analyzing the random combination of distillation of LLM and manual annotation (Kang et al., 2023) as well as active labeling assignments via the logit outputs (Wang et al., 2021). Our framework takes both quality and cost into consideration by using uncertainty metrics to make informed human-AI work-allocation decisions to ensure cost efficiency without compromising quality.

## 3 CoAnnotating Framework

Our *CoAnnotating* framework sets up a guide for annotating text data collaboratively (Figure 2). For a given unlabeled train dataset $D_t = \{t_1, t_2, ... t_m\}$ where $t_i$ is the i-th instance in the dataset, our framework automatically decides whether each data instance should be annotated by human or by the LLMs (Section 3.3) by computing the uncertainty level of the LLMs's annotations for each instance (Section 3.2), with the goal of achieving a higher annotation quality and a lower annotation cost for a given dataset (Section 3.4).

Text = Sentence1: {sentence1}
Sentence2: {sentence2}

| Prompt | Type |
|---|---|
| Please label if the following two sentences are paraphrases of each other. Please give your answer as "paraphrase" or "not paraphrase". {Text} | Instruction |
| {Text} Please label if the two sentences above are paraphrases of each other. Please give your answer as "paraphrase" or "not paraphrase". | Sequence Swapping |
| Given the following two sentences, please classify the relationship of the following two sentences as "paraphrase" or "not paraphrase". {Text} | Paraphrase |
| Is it true that the following two sentences are/are not paraphrases of each other? Give your answer as "true" or "false". {Text} | True/False |
| What relationship do the following two sentences have? Is it "paraphrase" or "not paraphrase"? {Text} | Question Answering |
| Please choose one option that best describes the relationship between the following two sentences. {Text} (A) Paraphrase (B) Not paraphrase | Multiple Choice Question |
| I think the following two sentences are/are not paraphrases of each other. Do you agree? {Text} | Question with Confirmation Bias |

Table 1: Examples of our 7 designed prompt types asking ChatGPT to annotate each instance for the concrete task of paraphrase detection.

## 3.1 Prompt Construction

Previous work shows that LLMs' performance can be highly sensitive to perturbations in input (Jang and Lukasiewicz, 2023). Therefore, we introduce a set of diverse types of prompts $P_i = \{p_{i1}, p_{i2}, ..., p_{ik}\}$ for each instance $t_i$. Besides the (1) basic instruction format, we vary the prompts by swapping its sequence of sentences (2; *symmetric perturbation*), paraphrasing the instruction (3; *semantic perturbation*), enquiring in various question formats (4; *True/False*, 5; *Textual Short Responses* 6; *Multiple Choice Question*) and asking with confirmation bias (7; *negation perturbation*).

## 3.2 Uncertainty Computation

In a real-world setting, there is no gold data on which to gauge the model's expected accuracy and thus decide on the optimal annotation strat-

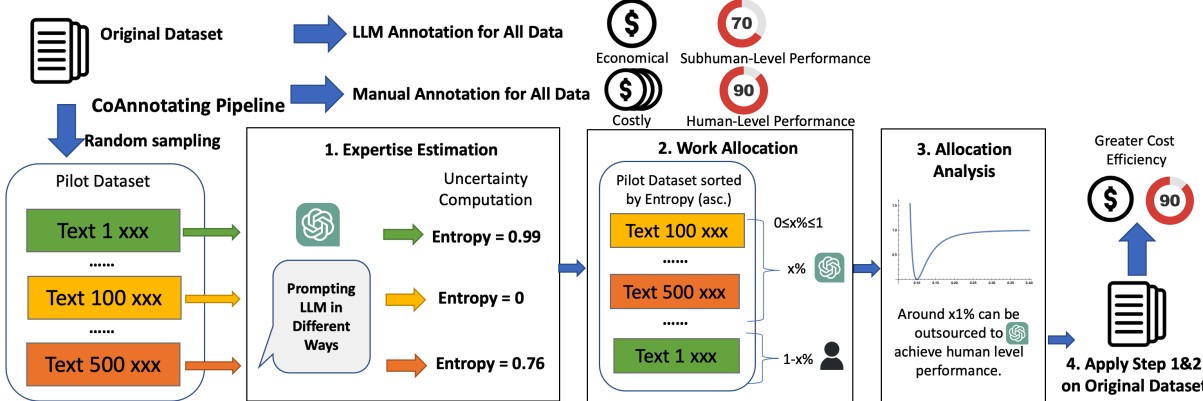

Figure 2: Workflow of *CoAnnotating*. The framework consists of uncertainty-guided expertise estimation, work allocation, and cost performance Pareto analysis. With insights gained from Pareto analysis on the pilot dataset, uncertainty-guided work allocation can be applied on the original unlabeled dataset to achieve greater cost efficiency.

egy. However, model confidence can serve as a reliable signal for model performance (Gentile et al., 2022; Diao et al., 2023; Wang et al., 2021). Therefore we compute the LLM uncertainty $u_i$ to guide the work-allocation process. We compute $u_i$ in two ways which are easy to implement and have proven effectiveness in previous literature (Diao et al., 2023): (1) self-evaluation and (2) entropy. In each case, for $t_i$ by prompting LLMs $k$ times with different prompts in $P_i$ we get $k$ annotations $A_i = \{a_{i1}, a_{i2}, ..., a_{ik}\}$ for each instance. As an ablation study (5.4), we also prompt LLMs k times with the same prompt to get $k$ annotations to study the effect of prompt perturbations.

**Self-Evaluation.** Previous work shows that LLMs are well calibrated and can provide information about their uncertainty themselves (Wang et al., 2021; Kadavath et al., 2022; Diao et al., 2023). We ask the model to directly output its confidence score (Lin et al., 2022) by postpending the phrase *"and please give a confidence score on a scale of 0 to 1 for your prediction"*. The uncertainty for $t_i$ is calculated by:

$$u_i = 1 - \frac{1}{k}\sum_{j=1}^{k} P_\theta(a_{ij}|p_{ij})$$

where $P_\theta(a_{ij}|p_{ij})$ is the probability of a class label being annotated by ChatGPT given the prompt $p_{ij}$. We obtain its value by extracting the confidence score provided by LLMs directly.

**Entropy.** Entropy is a measure of the impurity in a set of data and can be used to quantify the uncertainty associated with the class labels. The

larger the entropy value, the more uncertain the responses are. We can use this metric to estimate the uncertainty level:

$$u_i = -\sum_{j=1}^{k} P_\theta(a_{ij}|p_{ij})\ln P_\theta(a_{ij}|p_{ij})$$

where $P_\theta(a_{ij}|p_{ij})$ is calculated as the frequency of a certain prediction among all predictions.

### 3.3 Work Allocation Strategies

Building upon the aforementioned uncertainty level estimation, we can then use the uncertainty level $u_i$ to guide the work allocation.

**Random Allocation.** Random allocation is chosen as a baseline strategy for comparison. This is the strategy that randomly samples $n$ instances $(0 \le n \le m)$ in $D_t$ to be annotated by LLMs while the remaining $m - n$ data is annotated by humans.

**Self-Evaluation Guided Allocation.** Wang et al. (2021) introduces an active label assignment approach that ranks outputs by their logits. Not all LLM APIs support this computation, so we modify this baseline with our self-evaluation approach, sorting instances by the self-reported confidence scores in decreasing order. We then select the top $n$ instances $(0 \le n \le m)$ in $D_t$ with the lowest level of uncertainty as the best candidates for LLM annotation. The remaining $m - n$ data points are allocated to human annotators.

**Entropy Guided Allocation.** It is not possible to entirely ensure the reliability of black box LLMs self-reported confidence. Therefore, we also propose the use of entropy across LLMs' responses

to gauge their certainty and reliability. We sort the instances by their respective entropy values in increasing order and select the top $n$ instances ($0 \leq n \leq m$) in $D_t$ with the lowest level of uncertainty to be annotated by LLMs. Again, the remaining $m - n$ data points with inconsistent responses will be allocated for human annotation.

## 3.4 Strategy Selection

We frame the co-annotation process as a multi-objective optimization problem with two main objectives, maximizing annotation quality and minimizing annotation cost. We can determine annotation quality by the classification performance of a model fine-tuned using a certain co-annotation strategy. The total annotation cost is the sum of manual annotation costs and those incurred by the LLM. Inspired by Kang et al. (2023), we apply the Pareto efficiency concept in strategy selection. Here, the Pareto efficient scenario refers to the situation where it is impossible to increase the classification performance of the fine-tuned model without incurring a higher annotation cost. By adopting different allocation strategies and setting different proportions of data allocated to LLMs, we get various allocation patterns with different annotation qualities and costs. We can then plot the performances of each quality-cost combination and approximate the Pareto frontier by interpolating the discrete data points (Abdolrashidi et al., 2021; Treviso et al., 2022). Practitioners can plot annotation quality against the cost for pilot data to gain a better understanding of this tradeoff, and they can use the Pareto efficient points to decide which ratio of data they should outsource to LLMs at their desired budget level.

# 4 Experiments

## 4.1 Datasets

We use six classification datasets for different types of tasks. Since LLM inference costs much less than a human salary, we know the simple allocation decision is to choose LLMs over humans whenever an LLM achieves a utility greater than or equal to that of human annotators. For a more challenging setting, we identify tasks in which LLMs are known to struggle with discriminating the underlying constructs (Pikuliak, 2023; Wang et al., 2021). In such cases, there is a tradeoff between annotation quality and annotation cost and *CoAnnotating* facilitates better decision-making in such contexts. If the size

of the train data is too large, we will take a stratified random sampling for approximately 1000 samples.

**Topic Classification** is a challenging task for large pretrained language models like GPT-3 (Wang et al., 2021). We choose two representative datasets: TREC (Li and Roth, 2002) and AG News (Zhang et al., 2015). AG News contains news titles and their descriptions, which were gathered by an academic news search engine, and which span four topics: *world, sports, business, and science/technology.* TREC contains of English questions with six manually labeled class labels: *abbreviation; entity; description and abstract concept; human being; location;* and *numeric value.*

**Semantic Similarity** is known to challenge Chat-GPT (Jang and Lukasiewicz, 2023). We select MRPC (Dolan and Brockett, 2005) and TempoWiC (Loureiro et al., 2022) as two representative datasets for semantic similarity understanding. MRPC is a corpus of sentence pairs extracted from online news and annotated by humans for whether the sentences are semantically equivalent. TempoWiC contains annotated tweet pairs for whether there is a meaning shift of the target word.

**Nuanced Comprehension** We also experiment with Tweet Stance Detection (Mohammad et al., 2016a) and Conversation Gone Awry (Zhang et al., 2018) to explore the collaboration paradigm on tasks requiring more nuanced comprehension. Tweet Stance Detection in SemEval-2016 (Mohammad et al., 2016b) is a dataset of tweets annotated with the author's stance (favorable, neutral, and negative) toward a certain topic and we select the topic of abortion.

## 4.2 LLM Annotation

We obtain responses from ChatGPT (gpt-3.5-turbo) due to its high-quality annotations and low inference cost (Kuzman et al., 2023) using different prompts carefully crafted in Table 1. If the response is an ambiguous answer such as *"I cannot determine the class of the text"*, we encode it as a new class label which can result in higher uncertainty metrics. The uncertainty computation decides whether annotation will be finally allocated to ChatGPT, and if so, we decide the final label with a majority vote across ChatGPT's generations (Wang et al., 2022).

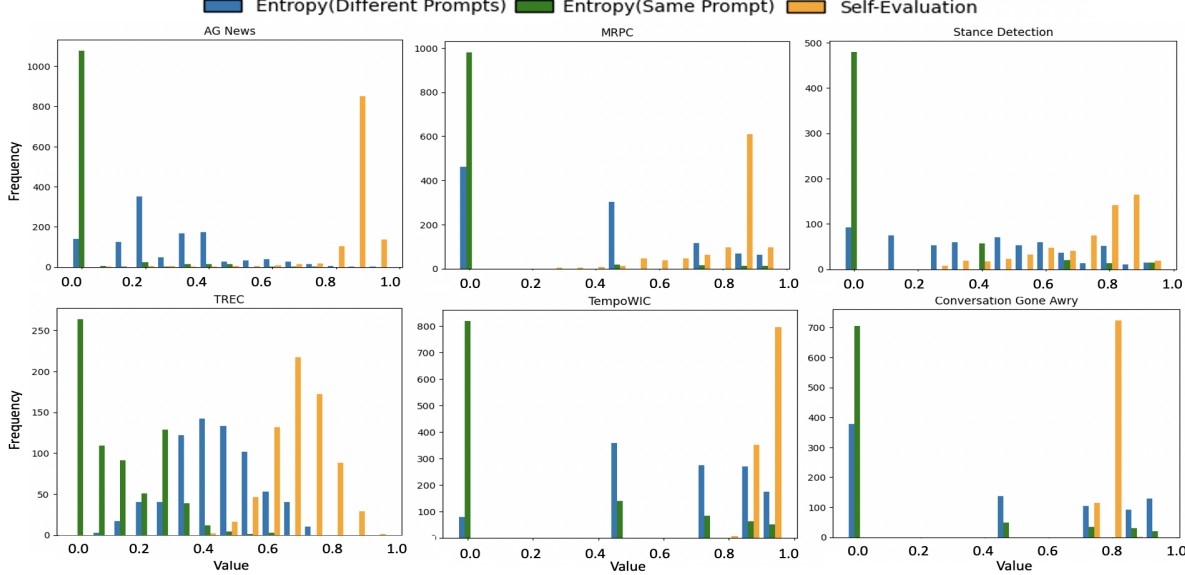

Figure 3: Distribution of entropy and confidence values.

## 4.3 Evaluation

To evaluate the quality of datasets annotated with different strategies, we fine-tune the same RoBERTa base classifier and calculate macro F1 scores on test data for a fair comparison. We report macro F1 as a more accurate representation of the performance due to the unbalanced nature of LLMs' annotations for some datasets.

In terms of cost, we only consider monetary cost in this work. We calculate human annotation costs based on what was reported in the dataset paper. If the information is not applicable, we assume each instance is annotated by 5 independent annotators with a wage of $15/hour. We calculate ChatGPT annotation cost using the product of the token length of the input prompt and the price of calling API for (`gpt-3.5-turbo`) ($0.002/1$k$ tokens) at the time of experimentation.

## 5 Results

### 5.1 Strategy Comparison

We plot the histograms for distribution of uncertainty metrics (entropy with different prompts and same prompt as well as confidence score). From Figure 3, we can observe that the model tends to be confident with its predictions with a skewed distribution towards high confidence value although we ask ChatGPT to normalize its answer.

We hypothesize that a lower level of uncertainty in ChatGPT's response indicates a higher degree of reliability in the label. Therefore, we set differ-

ent thresholds for entropy (lower than an entropy threshold) and self-confidence score (higher than a confidence threshold) to select data that ChatGPT is more certain about. For those instances selected, we evaluate ChatGPT's annotation quality by calculating its alignment with the gold label (human annotation). Figure 4's decreasing trends for entropy-guided allocation (green and blue dots) on all datasets validate our hypothesis of an inverse relationship between uncertainty and annotation quality. It justifies the helpfulness of using the entropy of ChatGPT's annotations as an estimate for its annotating expertise. Importantly, we observe that ChatGPT's self-reported confidence scores (orange dots) are not consistently a good estimate for its annotation quality. For some datasets such as AG News (top left), most of the data (94.3% with calculation) has high self-reported confidence ranging from 0.8 to 1, which leads to a weak separation of data in terms of annotation quality. For MRPC (top middle), there is a decreasing trend where data instances with higher confidence scores in fact have a poorer alignment with gold labels. This shows that the reliability of using self-reported confidence from LLMs is not guaranteed.

The purpose of achieving a higher quality for train data is to ensure that it can teach the classifier accurate information through fine-tuning. In Table 2, we carry out comparisons of different allocation strategies in terms of test performance after fine-tuning with such data labeled. We see that holding the proportion of data allocated to Chat-

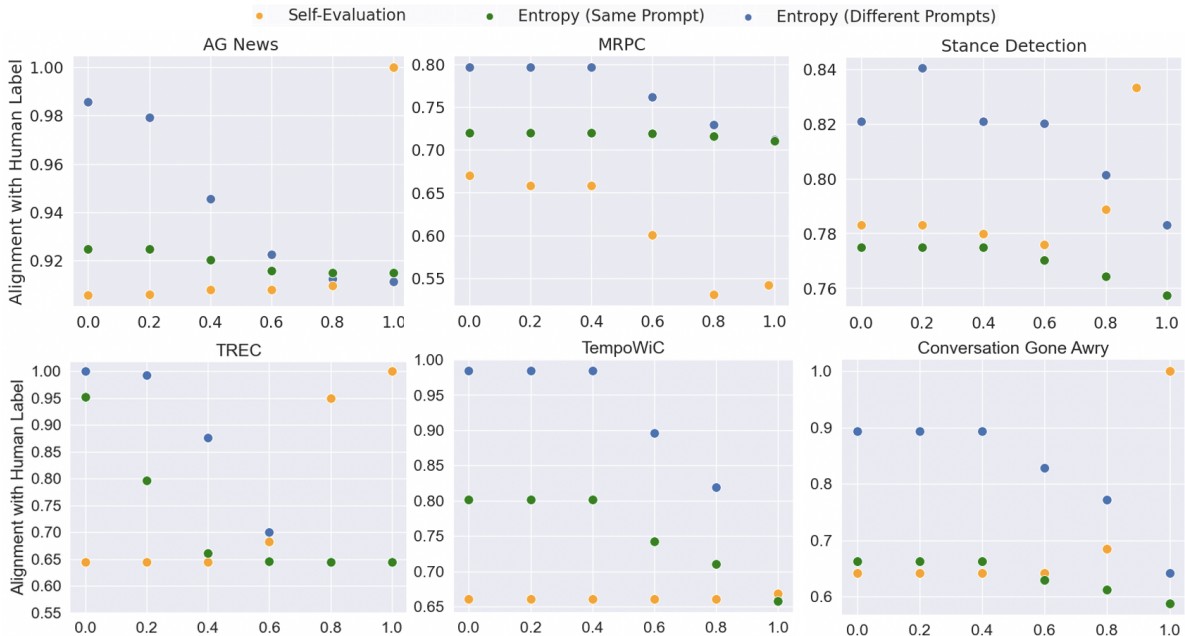

Figure 4: Scatter plots of the average alignment of ChatGPT's annotation with human annotation for train data against the threshold. We vary the threshold for different metrics during work allocation to investigate the effectiveness of different metrics in quantifying ChatGPT's annotation capability.

| | **AGNews** | | | | | | **TREC** | | | | | | **Stance Detection** | | | | | |
|---|---|---|---|---|---|---|---|---|---|---|---|---|---|---|---|---|---|---|
| **% ChatGPT** | 0 | 20 | 40 | 60 | 80 | 100 | 0 | 20 | 40 | 60 | 80 | 100 | 0 | 20 | 40 | 60 | 80 | 100 |
| **Strategies** | | | | | | | | | **Macro F1** | | | | | | | | | |
| Random | 88.2 | 87.9 | 85.8 | 79.8 | 81.8 | **82.6** | 92.1 | 88.1 | 86.1 | 81.6 | 76.4 | **75.8** | 60.2 | 53.9 | 53.6 | 55.0 | 50.4 | **53.6** |
| Self-Evaluation | 88.2 | 86.0 | 84.9 | 84.1* | 82.1 | 82.1 | 92.1 | 91.5 | 87.2 | **86.5*** | 76.4 | 74.3 | 60.2 | 56.9 | 54.8 | 54.4 | 52.8 | 52.9 |
| Entropy (Diff. Prompts) | 88.2 | **88.4** | **88.2** | **87.4*** | 84.0 | **82.6** | 92.1 | **91.9** | **87.4** | 80.8 | **79.2** | **75.8** | 60.2 | **58.2** | **55.1** | **56.8** | **54.7** | **53.6** |
| Entropy (Same Prompt) | 88.2 | 85.1 | 85.5 | 85.4* | **84.7** | 81.4 | 92.1 | 90.8 | 87.1 | 83.7 | 76.2 | 74.0 | 60.2 | 54.2 | 53.3 | 54.0 | 52.1 | 47.8 |

| | **TempoWIC** | | | | | | **MRPC** | | | | | | **Conversation** | | | | | |
|---|---|---|---|---|---|---|---|---|---|---|---|---|---|---|---|---|---|---|
| **% ChatGPT** | 0 | 20 | 40 | 60 | 80 | 100 | 0 | 20 | 40 | 60 | 80 | 100 | 0 | 20 | 40 | 60 | 80 | 100 |
| **Strategies** | | | | | | | | | **Macro F1** | | | | | | | | | |
| Random | 57.5 | 55.9 | 53.2 | 46.2 | 50.3 | 42.0 | 83.4 | 78.6 | 74.4 | 70.3 | 65.8 | **65.9** | 71.3 | 63.1 | 54.5 | 57.1 | 50.0 | 54.1 |
| Self-Evaluation | 57.5 | 57.8 | 55.9* | 51.8* | 52.9* | **43.0** | 83.4 | 79.7 | 77.8 | 71.5 | 63.9 | 58.6 | 71.3 | **70.1*** | 62.6 | **64.2*** | 50.4 | 50.7 |
| Entropy (Diff. Prompts) | 57.5 | **58.4*** | **56.9*** | **55.9*** | **53.8*** | 42.0 | 83.4 | **80.0** | **79.8*** | **76.6*** | **73.1*** | **65.9** | 71.3 | 66.5 | **64.2*** | 62.6 | **56.2*** | 54.1 |
| Entropy (Same Prompt) | 57.5 | 56.3 | 53.5 | 52.9* | 43.8 | 42.0 | 83.4 | 79.2 | 72.7 | 67.7 | 68.6 | 65.7 | 71.3 | 55.4 | 55.4 | 54.1 | 54.8 | **54.6** |

Table 2: Test performance of fine-tuned RoBERTa under different allocation strategies. We vary the percentage of data allocated to ChatGPT for annotation and carry out finetuning using train data annotated under different strategies for all six datasets. Figure with superscript * means the result under that strategy is significantly better than baseline strategy at 10% significance level.

GPT fixed (e.g., taking the setup of 40% for Tem-poWIC as an example), our proposed uncertainty-guided allocation using self-evaluation and entropy results in a better-annotated dataset, reflected by its higher test F1 (56.9) than the random allocation baseline (53.2). More often than not, entropy-guided allocation is better than confidence-guided allocation. This is probably due to the skewed distribution of self-reported confidence, resulting in a

poorer distinguishability between instances LLMs are better or worse at annotating.

## 5.2 Pareto Efficient Allocation

By plotting test performance against annotation cost (Figure 5), practitioners can visualize the trade-off in annotation quality achievable at different budgets with collaboration between human and an LLM like ChatGPT by studying the Pareto frontier.

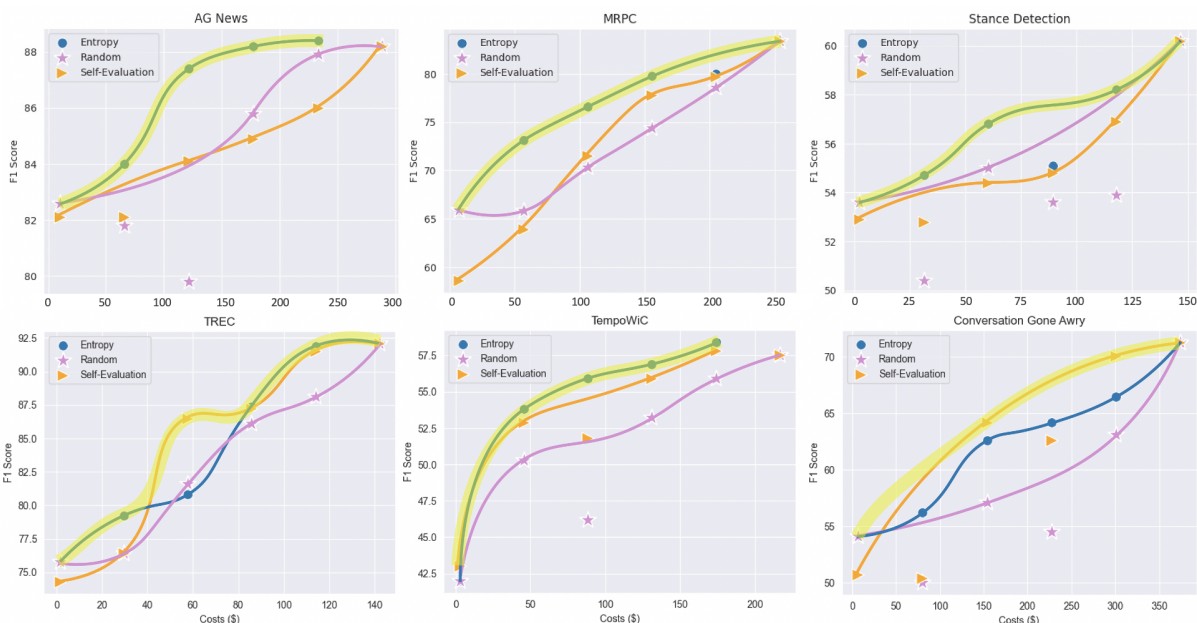

Figure 5: Pareto curves under different allocation strategies (random, entropy guided, self-evaluation guided). The Pareto frontier is highlighted, illustrating the optimal choices that are Pareto efficient.

| Dataset | Text | Groundtruth | ChatGPT |
|---|---|---|---|
| AG News | Title: Sprint Set to Debut Video-Streaming Cell Phone Description: OVERLAND PARK, Kan. (AP) – Channel surfing is moving off the couch as Sprint Corp... | Sci/Tech | Business |
| TREC | What does A&W of root beer fame stand for? | Abbreviation | Entity |
| Stance Detection | @user As a former fetus I oppose #ProlifeYouth #SemST | Negative | Neutral |
| Conversation | **rjoccolenty**: Shouldn't her name be Zainab Yusef and not Zainab Khan? **Bluebolt94**: Does the credits at the end of the episode say "Zainab Yusef"? No they say "Zainab Khan" and Yusef called her "Mrs. Khan" during the episode. So no, her name is "Zainab Khan". – **AnemoneProjectors**: The Khans are clearly not as traditional as the Masoods, or Afia would have been called Afia Yusef. We already know this! And what GS said. Watch the show properly P — | True | False |
| MRPC | **Sentence1:** At 5 p.m. EDT , Henri had maximum sustained winds near 50 mph , with some gusts reaching 60 mph. **Sentence2:** At 8 p.m. Friday , Henri was becoming disorganized , but still had maximum sustained winds near 50 mph , with stronger gusts. | Not paraphrase | Paraphrase |
| TempoWiC | **tweet 1**: If you need some to watch on Netflix, containment is so good. **tweet 2**: I have a lot of questions about the containment series. **target word**: containment | Same | Different |

Table 3: Specific instances with high entropy values for ChatGPT annotations.

Points along the highlighted Pareto frontier mean it is theoretically impossible to achieve a higher test accuracy without increasing the budget, and it is also impossible to reduce the cost but achieve the same level of annotation quality. Furthermore, it provides information on the approximate proportion that can be outsourced to ChatGPT to achieve human-level performance. For more straightforward tasks like topic classification, part of the annotation work could be potentially outsourced to ChatGPT and lead to a cost reduction (e.g., AG News: 33%) by ensuring human-level annotation performance. For datasets requiring nuanced comprehensions like Stance Detection and Conversa-

tion Gone Awry, any level of outsourcing to the current version of ChatGPT compromises annotation quality. Practitioners can choose among the Pareto efficient points based on their budgets.

### 5.3 Qualitative Analysis

We select some instances with entropy values higher than 0.8 from each dataset (Table 3) to understand the current challenges faced by ChatGPT in annotating data. We find that ChatGPT has high uncertainty for instances containing sarcasm and incomplete sentences that require more inference during opinion mining. For example, in deciding the stance towards abortion for the tweet "*as a for-*

*mer fetus I oppose*", the incomplete nature of this sentence causes confusion to ChatGPT. Also, it struggles with numerical reasoning as seen from its inability to compare wind speed during paraphrase detection and may be misled by some keywords ("Corp") related to other incorrect classes ("business") in topic classification.

### 5.4 Ablation Study

We carry out inferences with the same instruction-formatted prompt for the same number of times and compute the entropy for ChatGPT's responses. From Figure 4, we observe some extent of effectiveness of computing entropy using the same prompt in quantifying ChatGPT's capability, as reflected by a decreasing pattern of alignment with the increased threshold. However, it serves as a much weaker method to quantify expertise compared with our method with different prompt designs since the majority of the data has zero entropy (see Figure 3). This suggests that ChatGPT's responses are generally consistent within multiple applications of the same prompt. In Table 2, the test performance of entropy-guided allocation under different prompts is consistently higher than when based on a single prompt. The performance gap gives strong evidence of the utility of applying different prompt types in Table 1.

## 6 Conclusion

This work introduces *CoAnnotating*, a framework which takes a collaborative angle to view the relationship between humans and LLMs when annotating each dataset. Under this framework, we use uncertainty metrics to estimate LLMs' annotating capability and guide effective work allocation. Moreover, we apply the Pareto efficiency concept for practitioners to compare strategies and understand cost-performance tradeoffs. The empirical results demonstrate the effectiveness of our proposed framework in achieving greater cost efficiency. Overall, our framework provides important insights around the reliability of self-reported confidence score by LLMs, the sensitivity of ChatGPT's responses to prompt variations as well as the extent to which human resources can be freed by LLMs to be put on more meaningful areas.

## 7 Limitations

Since LLMs has been trained on a large number of datasets, there may be data leakage issue

where LLMs has seen some datasets in our experiment, making entropy values obtained for LLMs' responses lower. As an initial exploration of the co-annotating concept, this work aims for human-level performance in annotating datasets. It does not consider the scope of superhuman-level performance where we treat human annotation in each dataset as gold labels. Future work can further investigate the instances where LLMs actually annotates better than humans. We consider annotating profiles of human and LLMs as two groups but this framework can be further enriched by taking variations within each group (expert, crowd workers, different LLMs) into considerations. More exploration can also be carried out to investigate how to design prompts in a way that can increase LLMs's annotating expertise so that more annotation work can be outsourced to LLMs for greater cost efficiency. Moreover, this work only did experiments for classification tasks and English datasets. However, the idea of *CoAnnotating* is generalizable to generation tasks and datasets in other languages as well, which are meaningful to study in future work.

## Ethical Statement

We are aware of the potential ethical concerns of using LLMs as potential labelers in the data annotation process in terms of the perpetuation of existing biases in LLMs. Since LLMs are trained on vast amounts of texts on the Internet, they can unavoidably incorporate the biases present in these data sources. Such biases could be under-representation of certain demographic groups, cultural stereotypes as well as linguistic biases. However, we believe that the benefit of proposing a collaborative co-annotation framework outweighs the potential risks related to the framework.

## Acknowledgement

We are thankful to the members of SALT Lab and WING Lab as well as anonymous EMNLP reviewers for their helpful feedback. Minzhi Li is supported by the A*STAR Computing and Information Science (ACIS) Scholarship. Caleb Ziems is supported by the NSF Graduate Research Fellowship under Grant No. DGE-2039655. We would like to acknowledge a grant from the Office of Naval Research to DY, and National Research Foundation, Singapore under its AI Singapore Programme (AISG Award No: AISG2-GC-2022-005).

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

## A Model Setting

**ChatGPT parameters**: `temperature` = 0.7; `max_tokens` = 800; `top_p` = 0.95; `frequency_penalty` = 0; `presence_penalty` = 0; `openai_api_version` = '2023-03-15-preview'

**RoBERTa base parameters**: Adam optimizer, `learning_rate=2e-5`, `correct_bias` = True

## B Prompts

In this section, we present specific prompts used to obtain annotations from ChatGPT for each dataset during our experiments.

### B.1 AG News

**Instruction.** Please label the topic of the following news title and description as "world", "sports", "business" or "sci/tech".
Text: "Title: Wall St. Bears Claw Back Into the Black (Reuters) Description: Reuters Short-sellers, Wall Street's dwindling band of ultra-cynics, are seeing green again."

**Sequence Swapping.** Text: "Title: Wall St. Bears Claw Back Into the Black (Reuters) Description: Reuters - Short-sellers, Wall Street's dwindling band of ultra-cynics, are seeing green again." Please label the topic of the following news title and description as "world", "sports", "business" or "sci/tech".

**Paraphrasing.** Given the following text, please classify the topic of the following news title and description as "world", "sports", "business" or "sci/tech".
Text: "Title: Wall St. Bears Claw Back Into the Black (Reuters) Description: Reuters Short-sellers, Wall Street's dwindling band of ultra-cynics, are seeing green again."

**True/False 1.** Based on the language used, is it true that the following news title and description belongs to the topic of "world"?
Text: "Title: Wall St. Bears Claw Back Into the Black (Reuters) Description: Reuters Short-sellers, Wall Street's dwindling band of ultra-cynics, are seeing green again."

**True/False 2.** Based on the language used, is it true that the following news title and description belongs to the topic of "sports"?
Text: "Title: Wall St. Bears Claw Back Into the Black (Reuters) Description: Reuters Short-sellers,

Wall Street's dwindling band of ultra-cynics, are seeing green again."

**True/False 3.** Based on the language used, is it true that the following news title and description belongs to the topic of "business"?
Text: "Title: Wall St. Bears Claw Back Into the Black (Reuters) Description: Reuters Short-sellers, Wall Street's dwindling band of ultra-cynics, are seeing green again."

**True/False 4.** Based on the language used, is it true that the following news title and description belongs to the topic of "sci/tech"?
Text: "Title: Wall St. Bears Claw Back Into the Black (Reuters) Description: Reuters Short-sellers, Wall Street's dwindling band of ultra-cynics, are seeing green again."

**Question Answering.** What topic does the following news title and description belong to? Is it "world", "sports", "business" or "sci/tech"?
Text: "Title: Wall St. Bears Claw Back Into the Black (Reuters) Description: Reuters Short-sellers, Wall Street's dwindling band of ultra-cynics, are seeing green again."

**Multiple Choice Question.** Please choose one option that best describes the topic of the news title and description.
Text: "Title: Wall St. Bears Claw Back Into the Black (Reuters) Description: Reuters Short-sellers, Wall Street's dwindling band of ultra-cynics, are seeing green again."
Options:
(A)World
(B)Sports
(C)Business
(D)Sci/tech

**Confirmation Bias 1.** I think the following news title and description belongs to the "world" topic. Do you agree? Please give your answer as either yes or no.
Text: "Title: Wall St. Bears Claw Back Into the Black (Reuters) Description: Reuters Short-sellers, Wall Street's dwindling band of ultra-cynics, are seeing green again."

**Confirmation Bias 2.** I think the following news title and description belongs to the "sports" topic. Do you agree? Please give your answer as either yes or no.
Text: "Title: Wall St. Bears Claw Back Into the

Black (Reuters) Description: Reuters Short-sellers, Wall Street's dwindling band of ultra-cynics, are seeing green again."

**Confirmation Bias 3.** I think the following news title and description belongs to the "business" topic. Do you agree? Please give your answer as either yes or no.
Text: "Title: Wall St. Bears Claw Back Into the Black (Reuters) Description: Reuters Short-sellers, Wall Street's dwindling band of ultra-cynics, are seeing green again."

**Confirmation Bias 4.** I think the following news title and description belongs to the "sci/tech" topic. Do you agree? Please give your answer as either yes or no.
Text: "Title: Wall St. Bears Claw Back Into the Black (Reuters) Description: Reuters Short-sellers, Wall Street's dwindling band of ultra-cynics, are seeing green again."

## B.2   TREC

**Instruction.** Please label the type of the following question as "abbreviation", "entity", "description and abstract concept", "human being", "location", or "numeric value".
Text: "What makes a clitoris sensitive?"

**Sequence Swapping.** Text: "What makes a clitoris sensitive?"
Please label the type of the following question as "abbreviation", "entity", "description and abstract concept", "human being", "location", or "numeric value".

**Paraphrasing.** Given the following question, please classify the type of the following question as "abbreviation", "entity", "description and abstract concept", "human being", "location", or "numeric value".
Text: "What makes a clitoris sensitive?"

**True/False 1.** Based on the language used, is it true that the following question belongs to the class of abbreviation? Please give your answer as either "true" or "false".
Text: "What makes a clitoris sensitive?"

**True/False 2.** Based on the language used, is it true that the following question belongs to the class of entity? Please give your answer as either "true" or "false".
Text: "What makes a clitoris sensitive?"

**True/False 3.** Based on the language used, is it true that the following question belongs to the class of description and abstract concept? Please give your answer as either "true" or "false".
Text: "What makes a clitoris sensitive?"

**True/False 4.** Based on the language used, is it true that the following question belongs to the class of human being? Please give your answer as either "true" or "false".
Text: "What makes a clitoris sensitive?"

**True/False 5.** Based on the language used, is it true that the following question belongs to the class of location? Please give your answer as either "true" or "false".
Text: "What makes a clitoris sensitive?"

**True/False 6.** Based on the language used, is it true that the following question belongs to the class of numeric value? Please give your answer as either "true" or "false".
Text: "What makes a clitoris sensitive?"

**Question Answering.** Which class does the following question belong to? Is it "abbreviation", "entity", "description and abstract concept", "human being", "location", or "numeric value"?
Text: "What makes a clitoris sensitive?"

**Multiple Choice Question.** Please choose one option that best describes the class of the following question.
Text: "What makes a clitoris sensitive?"
Options:
(A)Abbreviation
(B)Entity
(C)Description and abstract concept
(D)Human being
(E)Location
(F)Numeric value

**Confirmation Bias 1.** I think the following question belongs to the abbreviation topic. Do you agree? Please give your answer as either "yes" or "no".
Text: "What makes a clitoris sensitive?"

**Confirmation Bias 2.** I think the following question belongs to the entity topic. Do you agree? Please give your answer as either "yes" or "no".
Text: "What makes a clitoris sensitive?"

**Confirmation Bias 3.** I think the following question belongs to the description and abstract concept

topic. Do you agree? Please give your answer as either "yes" or "no".
Text: "What makes a clitoris sensitive?"

**Confirmation Bias 4.** I think the following question belongs to the human being topic. Do you agree? Please give your answer as either "yes" or "no".
Text: "What makes a clitoris sensitive?"

**Confirmation Bias 5.** I think the following question belongs to the location topic. Do you agree? Please give your answer as either "yes" or "no".
Text: "What makes a clitoris sensitive?"

**Confirmation Bias 6.** I think the following question belongs to the numeric value topic. Do you agree? Please give your answer as either "yes" or "no".
Text: "What makes a clitoris sensitive?"

### B.3 MRPC

**Instruction.** Please label if the following two sentences are paraphrases of each other. Please give your answer as "paraphrase" or "not paraphrase".
Text: "Sentence1: " They are trying to turn him into a martyr , " said Vicki Saporta , president of the National Abortion Federation , which tracks abortion-related violence .
Sentence2: " We need to take these threats seriously , " said Vicki Saporta , president of the National Abortion Federation ."

**Sequence Swapping.** Text: "Sentence1: " They are trying to turn him into a martyr , " said Vicki Saporta , president of the National Abortion Federation , which tracks abortion-related violence .
Sentence2: " We need to take these threats seriously , " said Vicki Saporta , president of the National Abortion Federation ."
Please label if the following two sentences are paraphrases of each other. Please give your answer as "paraphrase" or "not paraphrase".

**Paraphrasing.** Given the following two sentences, please classify the relationship of the following two sentences as "paraphrase" or "not paraphrase".
Text: "Sentence1: " They are trying to turn him into a martyr , " said Vicki Saporta , president of the National Abortion Federation , which tracks abortion-related violence .
Sentence2: " We need to take these threats seriously

, " said Vicki Saporta , president of the National Abortion Federation ."

**True/False 1.** Is it true that the following two sentences are paraphrases of each other? Give your answer as "true" or "false".
Text: "Sentence1: " They are trying to turn him into a martyr , " said Vicki Saporta , president of the National Abortion Federation , which tracks abortion-related violence .
Sentence2: " We need to take these threats seriously , " said Vicki Saporta , president of the National Abortion Federation ."

**True/False 2.** Is it true that the following two sentences are not paraphrases of each other? Give your answer as "true" or "false".
Text: "Sentence1: " They are trying to turn him into a martyr , " said Vicki Saporta , president of the National Abortion Federation , which tracks abortion-related violence .
Sentence2: " We need to take these threats seriously , " said Vicki Saporta , president of the National Abortion Federation ."

**Question Answering.** What relationship do the following two sentences have? Is it "paraphrase" or "not paraphrase"?
Text: "Sentence1: " They are trying to turn him into a martyr , " said Vicki Saporta , president of the National Abortion Federation , which tracks abortion-related violence .
Sentence2: " We need to take these threats seriously , " said Vicki Saporta , president of the National Abortion Federation ."

**Multiple Choice Question.** Please choose one option that best describes the relationship between the following two sentences.
Text: "Sentence1: " They are trying to turn him into a martyr , " said Vicki Saporta , president of the National Abortion Federation , which tracks abortion-related violence .
Sentence2: " We need to take these threats seriously , " said Vicki Saporta , president of the National Abortion Federation ."
Options: (A)Paraphrase
(B)Not paraphrase

**Confirmation Bias 1.** I think the following two sentences are paraphrases of each other. Do you agree? Please give your answer in "yes" or "no".
Text: "Sentence1: " They are trying to turn him

into a martyr , ” said Vicki Saporta , president of the National Abortion Federation , which tracks abortion-related violence .
Sentence2: “ We need to take these threats seriously , ” said Vicki Saporta , president of the National Abortion Federation .”

**Confirmation Bias 2.** I think the following two sentences are not paraphrases of each other. Do you agree? Please give your answer in “yes” or “no”.
Text: “Sentence1: “ They are trying to turn him into a martyr , ” said Vicki Saporta , president of the National Abortion Federation , which tracks abortion-related violence .
Sentence2: “ We need to take these threats seriously , ” said Vicki Saporta , president of the National Abortion Federation .”

### B.4 TempoWiC

**Instruction.** Please label the meaning of the word frisk in the following 2 sentences as the same or different.
Sentence 1: imagine seeing qoute from cave story making it into smash as a dlc character instead of frisk or sans lmao
Sentence 2: Bloomberg? Are you people for real?16 cases of sexual harrassment, stop and frisk, redlining mortgages, and he is a conservative. If the dems think the only way you can beat trump is with a republican, then you deserve trump, and you show just how weak the dems are. #NotMeUs

**Sequence Swapping.** Sentence 1: imagine seeing qoute from cave story making it into smash as a dlc character instead of frisk or sans lmao Sentence 2: Bloomberg? Are you people for real?16 cases of sexual harrassment, stop and frisk, redlining mortgages, and he is a conservative. If the dems think the only way you can beat trump is with a republican, then you deserve trump, and you show just how weak the dems are. #NotMeUs
Please label the meaning of the word frisk in the following 2 sentences as the same or different.

**Paraphrasing.** Given the following text, please classify if the meaning of the word frisk in the following 2 sentences is the same or different.
Sentence 1: imagine seeing qoute from cave story making it into smash as a dlc character instead of frisk or sans lmao
Sentence 2: Bloomberg? Are you people for real?16 cases of sexual harrassment, stop and frisk,

redlining mortgages, and he is a conservative. If the dems think the only way you can beat trump is with a republican, then you deserve trump, and you show just how weak the dems are. #NotMeUs

**True/False 1.** Is it true that the meaning of the word frisk in the following 2 sentences is the same? Please give your answer in yes or no.
Sentence 1: imagine seeing qoute from cave story making it into smash as a dlc character instead of frisk or sans lmao
Sentence 2: Bloomberg? Are you people for real?16 cases of sexual harrassment, stop and frisk, redlining mortgages, and he is a conservative. If the dems think the only way you can beat trump is with a republican, then you deserve trump, and you show just how weak the dems are. #NotMeUs

**True/False 2.** Is it true that the meaning of the word frisk in the following 2 sentences is different? Please give your answer as yes or no.
Sentence 1: imagine seeing qoute from cave story making it into smash as a dlc character instead of frisk or sans lmao
Sentence 2: Bloomberg? Are you people for real?16 cases of sexual harrassment, stop and frisk, redlining mortgages, and he is a conservative. If the dems think the only way you can beat trump is with a republican, then you deserve trump, and you show just how weak the dems are. #NotMeUs

**Question Answering.** Considering the meaning of the word frisk in the following 2 sentences, is it the same or different? Please give your answer in “same” or “different”. Sentence 1: imagine seeing qoute from cave story making it into smash as a dlc character instead of frisk or sans lmao
Sentence 2: Bloomberg? Are you people for real?16 cases of sexual harrassment, stop and frisk, redlining mortgages, and he is a conservative. If the dems think the only way you can beat trump is with a republican, then you deserve trump, and you show just how weak the dems are. #NotMeUs

**Multiple Choice Question.** Please choose one option that best describes the meaning of the word frisk in the following 2 sentences:
Sentence 1: imagine seeing qoute from cave story making it into smash as a dlc character instead of frisk or sans lmao
Sentence 2: Bloomberg? Are you people for real?16 cases of sexual harrassment, stop and frisk, redlining mortgages, and he is a conservative. If

the dems think the only way you can beat trump is with a republican, then you deserve trump, and you show just how weak the dems are. #NotMeUs
(A) Same
(B) Different

**Confirmation Bias 1.** I think the meaning of the word frisk in the following 2 sentences is the same, do you agree?
Sentence 1: imagine seeing qoute from cave story making it into smash as a dlc character instead of frisk or sans lmao
Sentence 2: Bloomberg? Are you people for real?16 cases of sexual harrassment, stop and frisk, redlining mortgages, and he is a conservative. If the dems think the only way you can beat trump is with a republican, then you deserve trump, and you show just how weak the dems are. #NotMeUs

**Confirmation Bias 2.** I think the meaning of the word frisk in the following 2 sentences is different, do you agree?
Sentence 1: imagine seeing qoute from cave story making it into smash as a dlc character instead of frisk or sans lmao
Sentence 2: Bloomberg? Are you people for real?16 cases of sexual harrassment, stop and frisk, redlining mortgages, and he is a conservative. If the dems think the only way you can beat trump is with a republican, then you deserve trump, and you show just how weak the dems are. #NotMeUs

## B.5 Stance Detection

**Instruction.** Please label the stance towards abortion of the following text as "favourable", "neutral" or "negative".
Text: "we remind ourselves that love means to be willing to give until it hurts - Mother Teresa"

**Sequence Swapping.** Text: "we remind ourselves that love means to be willing to give until it hurts - Mother Teresa"
Please label the stance towards abortion of the following text as "favourable", "neutral" or "negative".

**Paraphrasing.** Given the following text, please classify the position towards abortion demonstrated in the following text as "favourable", "neutral" or "negative".
Text: "we remind ourselves that love means to be willing to give until it hurts - Mother Teresa"

**True/False 1.** Based on the language used, determine whether the following text holds a favourable stance towards abortion. Please give your answer as either "true" or "false".
Text: "we remind ourselves that love means to be willing to give until it hurts - Mother Teresa"

**True/False 2.** Based on the language used, determine whether the following text holds a neutral stance towards abortion. Please give your answer as either "true" or "false".
Text: "we remind ourselves that love means to be willing to give until it hurts - Mother Teresa"

**True/False 3.** Based on the language used, determine whether the following text holds a negative stance towards abortion. Please give your answer as either "true" or "false".
Text: "we remind ourselves that love means to be willing to give until it hurts - Mother Teresa"

**Question Answering.** What type of stance towards abortion is expressed in the following text? Is it favourable, neutral or negative?
Text: "we remind ourselves that love means to be willing to give until it hurts - Mother Teresa"

**Multiple Choice Question.** Please choose one option that best describes the stance towards abortion of the text.
Text: "we remind ourselves that love means to be willing to give until it hurts - Mother Teresa"
Options:
(A) Favourable
(B)Neutral
(C)Negative

**Confirmation Bias 1.** I think the following text has a favourable stance towards abortion. Do you agree? Please give your answer as either "yes" or "no".
Text: "we remind ourselves that love means to be willing to give until it hurts - Mother Teresa"

**Confirmation Bias 2.** I think the following text has a neutral stance towards abortion. Do you agree? Please give your answer as either "yes" or "no".
Text: "we remind ourselves that love means to be willing to give until it hurts - Mother Teresa"

**Confirmation Bias 3.** I think the following text has a negative stance towards abortion. Do you agree? Please give your answer as either "yes" or "no".

Text: "we remind ourselves that love means to be willing to give until it hurts - Mother Teresa"

## B.6 Conversation Gone Awry

**Instruction.** Please label if the conversation below will eventually derail into a personal attack. Even if you are uncertain, you must pick either "True" or "False" without using any other words.
Can.u.spel: im sorry to have been mistaken but whoever wrote that article made a miscalculation when determining the speed from knots to kph, 1 knot is 1.85 of a mile. therefore 10 knots is 18.5 miles per hour. am i not correct?
Greswik: You don't seem to know the difference between mph and kph. 25 knots is 46 km/h, as the article read already.

**Sequence Swapping.** Can.u.spel: im sorry to have been mistaken but whoever wrote that article made a miscalculation when determining the speed from knots to kph, 1 knot is 1.85 of a mile. therefore 10 knots is 18.5 miles per hour. am i not correct?
Greswik: You don't seem to know the difference between mph and kph. 25 knots is 46 km/h, as the article read already.
Please label if the conversation above will eventually derail into a personal attack. Even if you are uncertain, you must pick either "True" or "False" without using any other words.

**Paraphrasing** Given the following conversation, please classify if the conversation below will eventually derail into hate speech. Even if you are uncertain, you must pick either "True" or "False" without using any other words.
Can.u.spel: im sorry to have been mistaken but whoever wrote that article made a miscalculation when determining the speed from knots to kph, 1 knot is 1.85 of a mile. therefore 10 knots is 18.5 miles per hour. am i not correct?
Greswik: You don't seem to know the difference between mph and kph. 25 knots is 46 km/h, as the article read already.

**True/False 1.** Is it true that the conversation below will eventually derail into a personal attack? Even if you are uncertain, you must pick either "True" or "False" without using any other words.
Can.u.spel: im sorry to have been mistaken but whoever wrote that article made a miscalculation when determining the speed from knots to kph, 1 knot is 1.85 of a mile. therefore 10 knots is 18.5

miles per hour. am i not correct?
Greswik: You don't seem to know the difference between mph and kph. 25 knots is 46 km/h, as the article read already.

**True/False 2.** Is it true that the conversation below will not eventually derail into a personal attack? Even if you are uncertain, you must pick either "True" or "False" without using any other words.
Can.u.spel: im sorry to have been mistaken but whoever wrote that article made a miscalculation when determining the speed from knots to kph, 1 knot is 1.85 of a mile. therefore 10 knots is 18.5 miles per hour. am i not correct?
Greswik: You don't seem to know the difference between mph and kph. 25 knots is 46 km/h, as the article read already.

**Question Answering.** Considering the conversation below, will it eventually derail into a personal attack? Even if you are uncertain, you must pick either "True" or "False" without using any other words.
Can.u.spel: im sorry to have been mistaken but whoever wrote that article made a miscalculation when determining the speed from knots to kph, 1 knot is 1.85 of a mile. therefore 10 knots is 18.5 miles per hour. am i not correct?
Greswik: You don't seem to know the difference between mph and kph. 25 knots is 46 km/h, as the article read already.

**Multiple Choice Question.** Please choose one option that best describes the possible continuation of the following conversation. Can.u.spel: im sorry to have been mistaken but whoever wrote that article made a miscalculation when determining the speed from knots to kph, 1 knot is 1.85 of a mile. therefore 10 knots is 18.5 miles per hour. am i not correct? Greswik: You don't seem to know the difference between mph and kph. 25 knots is 46 km/h, as the article read already.
(A) it will eventually derail into a personal attack
(B) it will not eventually derail into a personal attack

**Confirmation Bias 1.** I think the conversation below will eventually derail into a personal attack, do you agree? Please give your answer in yes or no.
Can.u.spel: im sorry to have been mistaken but whoever wrote that article made a miscalculation

when determining the speed from knots to kph, 1 knot is 1.85 of a mile. therefore 10 knots is 18.5 miles per hour. am i not correct?

Greswik: You don't seem to know the difference between mph and kph. 25 knots is 46 km/h, as the article read already.

**Confirmation Bias 2.** I think the conversation below will not eventually derail into a personal attack, do you agree? Please give your answer in yes or no.

Can.u.spel: im sorry to have been mistaken but whoever wrote that article made a miscalculation when determining the speed from knots to kph, 1 knot is 1.85 of a mile. therefore 10 knots is 18.5 miles per hour. am i not correct?

Greswik: You don't seem to know the difference between mph and kph. 25 knots is 46 km/h, as the article read already.