# OpenReview forum: "CoAnnotating: Uncertainty-Guided Work Allocation between Human and Large Language Models for Data Annotation"
_EMNLP/2023/Conference — EMNLP 2023 Main_

### Official Review · Reviewer_HARE · 2023-08-04

**Soundness:** 3

**Excitement:**

4: Strong: This paper deepens the understanding of some phenomenon or lowers the barriers to an existing research direction.

**Paper Topic And Main Contributions:**

The paper introduces CoAnnotating, a novel paradigm for human and large language model collaboration in annotating unstructured text at scale. The framework uses uncertainty metrics to estimate LLMs' annotation capability and guide effective work allocation between humans and LLMs. CoAnnotating aims to achieve higher annotation quality and lower annotation cost for a given dataset. The authors use diverse prompts to compute uncertainty and explore various work allocation strategies, such as random allocation, self-evaluation guided allocation, and entropy guided allocation. Experiments were conducted on six classification datasets, and results demonstrate the effectiveness of the CoAnnotating framework in achieving greater cost efficiency. The study also provides insights into the reliability of self-reported confidence scores by LLMs and the sensitivity of LLM responses to prompt variations. While the framework focuses on human-level performance and classification tasks in English, the concept is generalizable to generation tasks and datasets in other languages, offering potential for future research.

**Questions For The Authors:**

1- How do you ensure that the quality of annotations is not compromised while focusing on cost reduction in the CoAnnotating framework?
2- What is the rationale behind the selection of the work allocation strategies presented in the paper, and can you provide a more detailed comparison of their performance?
3- How does the CoAnnotating framework perform when considering other LLMs or smaller models? Would the same uncertainty metrics be applicable?
4- Are there any specific domains or applications where the CoAnnotating framework may be particularly effective?

**Reasons To Accept:**

The authors propose a compelling approach to address the challenges in finding a balance between annotation quality, cost, and scalability. The framework is well-motivated and addresses a significant gap in the literature, as limited work has been done to leverage LLMs as complementary annotators and explore the allocation of annotation work between humans and LLMs. The paper should be accepted due to following reasons:
1- The CoAnnotating framework introduces a new paradigm in the annotation process by utilizing uncertainty metrics to estimate LLMs annotation capability. This enables an effective work allocation between humans and LLMs, which can help practitioners to achieve higher annotation quality and lower annotation costs.
2- The authors conduct experiments on six classification datasets, covering a wide range of tasks and difficulties. These experiments provide a robust evaluation of the proposed framework and demonstrate its effectiveness in various scenarios.
3- The authors compare the performance of their proposed framework with several baseline allocation strategies, such as random allocation, self-evaluation guided allocation, and entropy guided allocation. This thorough comparison showcases the advantages of the CoAnnotating approach over existing methods.
4- The paper offers an in-depth analysis of the results, highlighting the reliability of self-reported confidence scores by LLMs, the sensitivity of LLM responses to prompt variations, and the extent to which human resources can be freed by LLMs. The authors also acknowledge the limitations of their work, providing a foundation for future research directions.
Overall, the paper is well-written, thoroughly evaluated, and provides valuable insights into the potential of human-LLM collaboration. I believe that the findings presented in this paper will have a considerable impact on the development of future annotation systems and contribute to the ongoing conversation around the role of LLMs in the annotation process.

**Reasons To Reject:**

While the paper proposes an innovative framework for collaborative annotation between humans and large language models, there are several concerns that warrant reconsideration and refinement before the paper can be accepted. I recommend revising the paper to address the following issues:
1- The paper's motivation and problem statement are not clearly defined. The authors should provide a more detailed background on the challenges faced in manual annotation and the potential benefits of using LLMs as complementary annotators to set the stage for their proposed framework.
2- The primary focus of the paper is on cost reduction through the collaboration of humans and LLMs. However, the quality of the annotations is equally important in many scenarios. The authors should provide a more balanced discussion of both cost and quality aspects in their framework.
3- The paper presents several work allocation strategies but its lacking a clear rationale for their selection and a detailed comparison of their performance. The authors should provide a more comprehensive analysis of the allocation strategies and their benefits and drawbacks.
4- The uncertainty computation methods used in the framework, namely self-evaluation and entropy, are not adequately explained. The authors should provide a more detailed description of these methods and discuss their effectiveness in the context of the proposed framework.
5- The evaluation of the framework is limited to six classification datasets. To demonstrate the robustness and generalizability of the proposed framework, the authors should consider extending their evaluation to other tasks, such as sequence tagging, and a wider range of datasets.
6-	The authors do not provide sufficient ablation studies to understand the individual contributions of different components of the framework. A deeper analysis of the various aspects of the framework, such as the impact of different prompt designs and uncertainty computation methods, would strengthen the paper.
Overall, the CoAnnotating framework presents a promising approach to human-LLM collaboration in annotating unstructured text. However, the paper requires significant revisions to address the concerns mentioned above and provide a more comprehensive and well-rounded evaluation. I recommend revising the paper to address these issues.


**Reproducibility:**

4: Could mostly reproduce the results, but there may be some variation because of sample variance or minor variations in their interpretation of the protocol or method.

**Reviewer Confidence:**

5: Positive that my evaluation is correct. I read the paper very carefully and I am very familiar with related work.

**Typos Grammar Style And Presentation Improvements:**

There is typo in page 4 Section 3.3 first line: "Building upon the aforementioned uncertainty level estimation, we can then such uncertainty level ui to guide the work allocation." Replace "such" with perhaps "use".

---

> ### Author Rebuttal · Authors · 2023-08-28
>
> Thank you for your thoughtful reviews and acknowledgement of our work as one which addresses a significant research gap with thorough analysis and interesting insights.
>
> ### Response to ‘Reasons to Reject’:
> 1) **Challenges in manual annotation:** Some challenges in manual annotation include subjectivity, annotation inconsistency, cost in terms of recruiting and training annotators. We will add these aspects in the camera-ready.
> 2) **Too much focus on cost reduction:** In CoAnnotating framework, we did not prioritize one over the other. Instead, we identified both quality and cost as core objectives. As illustrated in Section 3.4 and Figure 4, we set up a multi-objective optimization process which aims to maximize annotation quality while minimizing annotation costs at the same time.
> 3) **Rationale in selecting the work allocation strategies:** We talked about the rationales in selecting the work allocation strategies in Section 3.2, which are (1) easy implementation and (2) proven effectiveness in previous literature. We made detailed comparisons for different strategies in Section 5.1, using the subsection talking about relative performance of different working allocations in terms of alignment with human label, annotation quality as well as annotation costs.
> 4) **Computational method explanation and discussion of their effectiveness:** We explained the computational method of self-evaluation and entropy calculation by defining mathematical variables and explaining formulae for each of them in Section 3.2. For more mathematical details, we will include them in the appendix. We also discussed their relative effectiveness through visualizations (Figure 3 and Figure 4) and words (Line 405-417, Line 435-441).
> 5) **Task selection:** To demonstrate the general capability, we selected tasks from a diversity of domains. We did not opt for sequence tagging tasks, as there exists some difficulty by directly applying the framework. It wouldn’t make sense to split the work at individual tags since both an LLM and a human annotator need to consider the whole sequence to provide the tag. And it might be difficult to compare performance at the level of sentences. If we need a distinct prompt for each tag, the length of the prompt would be variable and thus the cost of inference would be variable.
> 6) **Ablation study:** We actually included the ablation study to study the impact of different prompt designs in Section 5.4 and managed to show that the design of different prompts in our framework is effective in improving work allocation. We used the trend in Figure 3 and numerical results in Table 2 to support our claim by comparing performance with the same prompt and different prompts. For different uncertainty metrics, we did not include that in the main section because our main focus is to show that our framework using uncertainty metrics as a guide works better than previous random allocation methods. Comparing different uncertainty metrics is less of a focus here but it will be interesting to explore it in future work.
>
> ### Response to ‘Questions for Authors’:
> **Question 1:** As mentioned in Section 5.2 (Line 454-464), it is possible to reduce costs while ensuring human-level performance for more straightforward tasks like topic classification. On the other hand, it is still inevitable to compromise the annotation quality while cutting costs for tasks requiring nuanced comprehension. Practitioners are advised to choose among the Pareto efficient points based on their specific budgets. Therefore, the compromise is sometimes inevitable but the use of Pareto efficiency concept helps practitioners to achieve the highest possible annotation quality for a specific budget. \
> **Question 2:** We mentioned the rationales behind the selection of the work allocation strategies (1. Easy implementation [Bansiya et al., 1999] 2. Proven effectiveness in previous literature [Diao et al., 2023]) in Section 3.2. We made detailed comparisons in Section 5.1, talking about relative performance of different working allocations in terms of alignment with human label, annotation quality as well as annotation costs. \
> **Question 3:** Our CoAnnotating framework serves as a general framework that can be applied to any large language model with varied annotation costs and quality. The models may evolve over time but our framework can still hold. Practitioners can follow the same flow including computing the same uncertainty metrics for other models or new models that come out in future. \
> **Question 4:** Thanks for the thought-provoking question! The CoAnnotating framework can be particularly effective for tasks requiring expert annotation (e.g. finance, legal domain) but under constrained budgets. This is because expert annotations are more costly, which may make having all data annotated by humans impossible if the budget is limited. Our proposed framework can suggest the best allocation pattern using Pareto efficiency frontier and help practitioners achieve the highest possible annotation quality given a constrained budget.
>
> [Bansiya et al., 1999] Bansiya, J., Davis, C., & Etzkorn, L. (1999). An entropy‐based complexity measure for object‐oriented designs. Theory and practice of object systems, 5(2), 111-118.
>
> [Diao et al., 2023] Diao, S., Wang, P., Lin, Y., & Zhang, T. (2023). Active prompting with chain-of-thought for large language models. arXiv preprint arXiv:2302.12246.

---

### Official Review · Reviewer_Tt3q · 2023-08-07

**Soundness:** 5

**Excitement:**

4: Strong: This paper deepens the understanding of some phenomenon or lowers the barriers to an existing research direction.

**Paper Topic And Main Contributions:**

This paper proposes a new paradigm for LLM-human collaboration in annotation tasks since that topic has not been fully explored yet. Using model confidence as a proxy for model competence, it claims that this new method can achieve improved resource allocation where costs and quality are well-balanced. Since model confidence is not always reliable, the authors test different prompts and allocation strategies to achieve a more reliable signal.

**Questions For The Authors:**

A. Can you elaborate on how the Pareto frontier was calculated?

**Reasons To Accept:**

The topic of using LLMs as annotators is a very timely topic, and this paper does a very thorough job in its analysis. The analysis has enough content as a main contribution. Additionally, the paper is structured nicely and the visualizations are helpful for understanding the content.

**Reasons To Reject:**

There is a lot of content in this paper, and it might benefit from being split into a long and short paper. Since the Pareto efficiency work wasn't mentioned until the latter half of the paper, that could perhaps go into a shorter paper geared more towards an industry audience.

**Reproducibility:**

4: Could mostly reproduce the results, but there may be some variation because of sample variance or minor variations in their interpretation of the protocol or method.

**Reviewer Confidence:**

4: Quite sure. I tried to check the important points carefully. It's unlikely, though conceivable, that I missed something that should affect my ratings.

**Typos Grammar Style And Presentation Improvements:**

For Figure 3, the bracketed numbers are a little distracting. Could you bin the proportion of samples and use different marker shapes to indicate them or use marker size?

For Table 2, I personally think highlighting the cells would make the pattern easier to see in the table.

---

> ### Author Rebuttal · Authors · 2023-08-28
>
> Thank you for your thoughtful reviews and appreciation of the thorough analysis of our work.
>
> ### Response to ‘Reasons to Reject’:
> Thank you for your suggestion to split the work to a long and short paper. We included the Pareto efficiency concept to ensure the completeness of the whole pipeline. In the camera-ready, we will better integrate this section with the first half of the paper.
>
> ### Response to ‘Questions for Authors’:
> We plot the performances of each quality-cost combination and approximate the Pareto frontier by interpolating the discrete data points [Abdolrashidi et al., 2021] [Treviso et al., 2020]. We will add the method and relevant references to the camera-ready to make it clearer.
>
> [Abdolrashidi et al., 2021] Abdolrashidi, A., Wang, L., Agrawal, S., Malmaud, J., Rybakov, O., Leichner, C., & Lew, L. (2021). Pareto-optimal quantized resnet is mostly 4-bit. In Proceedings of the IEEE/CVF Conference on Computer Vision and Pattern Recognition (pp. 3091-3099).
>
> [Treviso et al., 2020] Treviso, M., Góis, A., Fernandes, P., Fonseca, E., & Martins, A. F. (2021). Predicting attention sparsity in transformers. arXiv preprint arXiv:2109.12188.

---

### Official Review · Reviewer_rzwv · 2023-08-08

**Typos Grammar Style And Presentation Improvements:** The format of Figure 3 and Table 2 ca…
**Soundness:** 3

**Excitement:**

4: Strong: This paper deepens the understanding of some phenomenon or lowers the barriers to an existing research direction.

**Paper Topic And Main Contributions:**

This paper focuses on data annotation challenges in NLP tasks, and proposes a co-annotation framework for human-AI collaboration during the annotation process. The framework aims to achieve higher annotation quality and lower annotation costs for a given dataset. It does this by automatically deciding whether each data instance should be annotated by a human or by LLM. Experiments using annotated data to fine-tune RoBERTa on multiple benchmark datasets demonstrate that the proposed method achieves Pareto efficient results in balancing cost and annotation quality.


**Questions For The Authors:**

1. Could it be considered to add some results of LLMs solving these problems under zero-shot and few-shot settings? It's possible that using ChatGPT itself may achieve better results than fine-tuning RoBERTa with annotated data.
2. Could evaluations on different LLMs be added? For example, using other LLMs, would there be some differences in cost and annotation quality?

**Reasons To Accept:**

1. The proposed annotation framework demonstrates considerable practical value in alleviating the lack of supervised data across various NLP tasks.
2. Comprehensive experiments conducted include annotating diverse datasets and fine-tuning a RoBERTa model for evaluation. These thorough experimental designs and results clearly showcase the superiority and cost-effectiveness of this novel approach.
3. The paper is well-written and well-structured.
4. The appendix is abundant and provides prompts suitable for various tasks.


**Reasons To Reject:**

1. Expanding the evaluation to include sequence labeling, NLI, and other diverse NLP tasks beyond just classification-based tasks would better demonstrate the general applicability of the proposed framework across different task types.
2. The proposed framework is not combined with active learning paradigm. Active learning is a very natural solution that can be used to improve labeling efficiency and reduce labeling costs when it comes to human-AI collaboration. The proposed annotation workflow is theoretically orthogonal to active learning.
3. The few-shot capability of LLMs is not utilized. It has been proven that LLMs can significantly improve performance given a small number of demonstrations. Perhaps in the framework proposed in this paper, using the human annotated data as demonstrations and inputting them into the LLM could further improve the results.

**Reproducibility:**

3: Could reproduce the results with some difficulty. The settings of parameters are underspecified or subjectively determined; the training/evaluation data are not widely available.

**Reviewer Confidence:**

3: Pretty sure, but there's a chance I missed something. Although I have a good feel for this area in general, I did not carefully check the paper's details, e.g., the math, experimental design, or novelty.

---

> ### Author Rebuttal · Authors · 2023-08-28
>
> Thank you for your thoughtful reviews and appreciation of the considerable practical value of our work.
>
> ### Response to ‘Reasons to Reject’:
> 1) **Task selection:** To demonstrate the general capability, we selected tasks from a diversity of domains. We did not opt for sequence tagging tasks, as there exists some difficulty by directly applying the framework. It wouldn’t make sense to split the work at individual tags since both an LLM and a human annotator need to consider the whole sequence to provide the tag. And it might be difficult to compare performance at the level of sentences. If we need a distinct prompt for each tag, the length of the prompt would be variable and thus the cost of inference would be variable. We also chose not to consider some traditional NLI tasks like MNLI and QNLI [LLM+NLU], as LLMs can already achieve a very high F1 score and there will be no significant gap between human labeling and LLM labeling. Data leakage may be one explanation for LLMs’ strong performance on NLI; future work can consider newer unleaked NLI datasets if they are released.
> 2) **Integration with active learning:** Thanks for pointing out active learning as one orthogonal area that our framework can be integrated with. We did not combine them in order to just study the effectiveness of our proposed framework, but it is a promising future extension to combine these two components to further improve the efficiency of the whole annotation pipeline.
> 3) **Testing few-shot capability:** We did not experiment with the few-shot capability of LLMs because few-shot settings require annotated data (usually by human). We want to best understand the orthogonal contributions of human annotation and LLM annotation by disentangling the two. It will be interesting for follow-up works to build upon our framework to understand their interactions.
>
> ### Response to ‘Questions for Authors’:
> **Question 1:** Here are some findings in previous work:
> 1) AGNews: GPT3-Shot1 achieves 0.74 accuracy [Wang et al., 2021] which is worse than fine-tuning on annotated data (0.94)
> 2) TREC: GPT3-Shot1 achieves 0.75 accuracy  [Wang et al., 2021] which is worse than fine-tuning on annotated data (0.95)
> 3) MRPC: ChatGPT achieves 0.72 F1 score [Zhong et al., 2023] which is worse than fine-tuning on annotated data (0.93)
> 4) TempoWIC: ChatGPT achieves 0.56 accuracy which is worse than fine-tuning on annotated data (0.66) [Ziems et al., 2023]
> 5) TweetStance: ChatGPT achieves 0.56 accuracy [Kocoń et al., 2023] which is worse than fine-tuning on annotated data (0.69)
> 6) Conversation Gone Awry: ChatGPT achieves 0.53 accuracy which is worse than fine-tuning on annotated data (0.65) [Ziems et al., 2023]
>
> **Question 2:** Yes, different models could be added. Although the empirical analysis is done for one representative of large language models - ChatGPT for a consistent comparison and analysis, we design CoAnnotating as a general framework that can be applied to any large language model with varied annotation cost and quality. This framework can still hold when new models with higher capability come out in future and we encourage future work to adopt our framework with different models.
>
> [Wang et al., 2021] Wang, S., Liu, Y., Xu, Y., Zhu, C., & Zeng, M. (2021). Want to reduce labeling cost? GPT-3 can help. arXiv preprint arXiv:2108.13487.
>
> [Kocoń et al., 2023] Kocoń, J., Cichecki, I., Kaszyca, O., Kochanek, M., Szydło, D., Baran, J., ... & Kazienko, P. (2023). ChatGPT: Jack of all trades, master of none. Information Fusion, 101861.
>
> [Zhong et al., 2023] Zhong, Q., Ding, L., Liu, J., Du, B., & Tao, D. (2023). Can chatgpt understand too? a comparative study on chatgpt and fine-tuned bert. arXiv preprint arXiv:2302.10198.
>
> [Ziems et al., 2023] Ziems, C., Held, W., Shaikh, O., Chen, J., Zhang, Z., & Yang, D. (2023). Can Large Language Models Transform Computational Social Science?. arXiv preprint arXiv:2305.03514.

---

### Official Review · Reviewer_mCqk · 2023-08-12

**Typos Grammar Style And Presentation Improvements:** No comments on style/grammer/presenta…
**Soundness:** 4

**Excitement:**

4: Strong: This paper deepens the understanding of some phenomenon or lowers the barriers to an existing research direction.

**Missing References:**

This work seems very related to active learning in general and I would have expected more discussion on that point.  A few examples of a very similar systems include:
Qiu, Hang, Krishna Chintalapudi, and Ramesh Govindan. "MCAL: Minimum Cost Human-Machine Active Labeling." The Eleventh International Conference on Learning Representations. 2022.  https://openreview.net/forum?id=1FxRPKrH8bw
Beluch, William H., et al. "The power of ensembles for active learning in image classification." Proceedings of the IEEE conference on computer vision and pattern recognition. 2018.
Established literature for active learning includes
 S. Tong. Active learning: theory and applications. Stanford University, 2001.

**Paper Topic And Main Contributions:**

A common problem in NLP is obtaining sufficiently high-quality training data at a reasonable cost.  Because LLMs have strong zero-shot performance and are much cheaper to use instead of paying a human's salary, many researchers may consider using them for annotation instead.  This paper proposes an alternative, the CoAnnotating Framework, which describes a set of techniques for choosing when to use LLM annotation and when to use the more expensive human annotation when the LLM is uncertain.  The paper discusses a few different methods for determining the uncertainty of the LLM: self-evaluation, where the model is asked to report its own uncertainty, entropy using samples from the same prompt, and entropy using samples from different prompts.  The different uncertainty methods are compared by looking at the performance of a downstream task-specific BERT model trained on the respective annotations.  Three tasks are considered: topic classification using TREC and AG News datasets, semantic similarity using MRPC dataset, and nuanced comprehension using the Tweet Stance Detection and Conversation gone awry datasets.  The methods are compared to a random allocation baseline.  Results show significant improvement for many of the tasks by using CoAnnotation Framework vs a random allocation, including many that are at a 10% significance level.  Additionally, they show how the methods compare to the optimal Pareto frontier in terms of F1/Cost($) tradeoff, where the methods closely track the frontier, especially as compared to random allocation.  The main limitations identified include potential leakage of test datasets with LLM training sets, lack of detailed breakdown of the annotation groups, prompt design, and how it is focused on English language annotation.  Ethical concerns identified include perpetuating bias in LLMs to the resulting datasets.

**Questions For The Authors:**

Question A) How are you computing the Pareto frontier in Figure 4? I think the reader needs to understand how that is being arrived at in order to give context to how close the proposed framework is to that frontier.

Question B) How would you extend the framework to take into account the inference cost of different quality LLMs?

Question C) GIven the large similarity to active learning in general, why not compare it to other active learning methods such as query-by-committee or density-based approaches?  Only comparing to random allocation seems a simple baseline to improve upon.

**Reasons To Accept:**

The paper identifies a timely problem, how to use LLMs as annotators effectively, and proposes and demonstrates a solution framework to address the LLM annotation problem.  The results convincingly show that the proposed framework is better than random allocation with many points showing 10% significance level.  They also present evidence showing it is also close to Pareto frontier for many of the datasets evaluated.

While uncertainty based methods have been studied in other work as a method of choice for getting a human label (see missing references), I think the results of uncertainty as applied to ChatGPT are in themselves interesting and have value.

**Reasons To Reject:**

While this paper is well done and it's possible applying these ideas to the latest LLMs like ChatGPT is novel, I'm still concerned that the approach is not very novel in general - please see papers in the missing references section related to active learning.  Using uncertainty to select which machine labeled samples to have humans label is a very established idea.  For example consider this paper:
Qiu, Hang, Krishna Chintalapudi, and Ramesh Govindan. "MCAL: Minimum Cost Human-Machine Active Labeling." The Eleventh International Conference on Learning Representations. 2022.  https://openreview.net/forum?id=1FxRPKrH8bw
That paper presents a very simlar system, but for CV applications.  This new application area is very relevant now and so the analysis is interesting, but my concern is if you replaced ChatGPT with a BERT or RNN or X older model the rest of the approach/framework seems very standard and well studied.

Related to above the only baseline compared to is random allocation, but there is a lot of other active learning strategies in literature such as query-by-committee and desnity-based approaches.  Having these more creative baselines may have been more interesting than the simple random allocation baseline.

A limitation not identified in the paper is that only a single LLM model was considered, but there are many choices for such an LLM and that choice is also based on cost (although not as much as human vs LLM).  For example choosing between ChatGPT and GPT-4 has a significant cost impact at current prices with quality tradeoff at large scale.  It would have been interesting to also have that choice in the framework, perhaps that's an obvious extension.

**Reproducibility:**

4: Could mostly reproduce the results, but there may be some variation because of sample variance or minor variations in their interpretation of the protocol or method.

**Reviewer Confidence:**

4: Quite sure. I tried to check the important points carefully. It's unlikely, though conceivable, that I missed something that should affect my ratings.

---

> ### Author Rebuttal · Authors · 2023-08-28
>
> Thank you for your thoughtful reviews and acknowledgement of our work as one which identifies a timely problem of how to use LLMs effectively.
>
> ### Response to ‘Reasons to Reject’:
> 1) **Difference between our work and active learning:** Our work is *“theoretically orthogonal to active learning”* (see Reviewer rzwv). Active learning is the process of selecting worthy training data which can provide the most information to the model to be labeled (usually by human annotators). It helps to save time and resources by reducing the number of examples to be labeled. On the other hand, our framework is an annotation allocation strategy for human-LLM teams working on a fixed set of data points to be labeled. It can be integrated with active learning but it is not a type of active learning approach. Using our experimental setting as one example, one can apply active learning approaches to select a subset of unlabeled data that is challenging to a RoBERTa model and then use our CoAnnotating framework to allocate the annotations between human labelers and large language models.
> 2) **Baseline selection:** As explained above, our work is not studying active learning strategies for large language models so we did not use traditional active learning strategies like query-by-committee and density-based approaches as baselines. The main research focus of our work is to tackle the problem of how to effectively assign annotation work to humans and large language models. Previous work aiming to solve this assignment problem only considered (1) random allocation [Kang et al., 2023] and (2) using LLMs’ self-reported confidence as allocation criteria [Wang et al., 2021], so we adopted these two methods as baselines. They are more suitable baselines since they cover the same research problem as our work.
> 3) **Consideration of different LLMs beyond ChatGPT:** Thank you for pointing out this interesting point for future work! We actually mentioned it in our Limitations section (line 532-536): *“We consider annotating profiles of human and LLMs as two groups but this framework can be further enriched by taking variations within each group (expert, crowdworkers, different LLMs) into considerations.”*
>
> ### Response to ‘Questions for Authors’:
> **Question A:** We plot the performances of each quality-cost combination and approximate the Pareto frontier by interpolating the discrete data points [Abdolrashidi et al., 2021] [Treviso et al., 2020]. We will add the method and its relevant references to the camera-ready to make it clearer. \
> **Question B:** Our methods of estimating LLMs’ expertise by uncertainty and cost computations can generalize to different LLMs. Here is one possible way our framework can be extended: we can still apply the uncertainty method proposed in our work to estimate the annotation expertise of different models and rank them based on entropy (ascending order) for each model. If an instance appears in the top k% of the sorted lists for several models, we will label it using the model with the lowest inference cost. If an instance does not appear in the top k% of the sorted list for any model, we will assign it to a human annotator. This is a straightforward extension of our current framework with minimal modifications and it can be further improved with other justified modifications in future work. \
> **Question C:** As explained in our response 2 to ‘Reasons to Reject’, our work is theoretically orthogonal to active learning so we did not set traditional active learning strategies like query-by-committee and density-based approaches as baselines. Strategies we selected as baselines (random allocation and using LLMs’ self-reported confidence) are existing previous methods which share the same research focus of our work so they are more suitable. However, we appreciate your suggestions and think other active strategies you suggested can inspire future extensions of our proposed methods to effectively allocate work between human and large language models.
>
>
> [Kang et al., 2023] Kang, J., Xu, W., & Ritter, A. (2023). Distill or annotate? cost-efficient fine-tuning of compact models. arXiv preprint arXiv:2305.01645.
>
> [Wang et al., 2021] Wang, S., Liu, Y., Xu, Y., Zhu, C., & Zeng, M. (2021). Want to reduce labeling cost? GPT-3 can help. arXiv preprint arXiv:2108.13487.
>
> [Abdolrashidi et al., 2021] Abdolrashidi, A., Wang, L., Agrawal, S., Malmaud, J., Rybakov, O., Leichner, C., & Lew, L. (2021). Pareto-optimal quantized resnet is mostly 4-bit. In Proceedings of the IEEE/CVF Conference on Computer Vision and Pattern Recognition (pp. 3091-3099).
>
> [Treviso et al., 2020] Treviso, M., Góis, A., Fernandes, P., Fonseca, E., & Martins, A. F. (2021). Predicting attention sparsity in transformers. arXiv preprint arXiv:2109.12188.

---

### Meta-Review · Area_Chair_syMx · 2023-09-15

**Recommendation:** 5

**Metareview:**

This paper proposes a framework for annotating data using both humans and LLMs. The core idea is to use the – cheaper – LLM to generate an annotation for examples where it has high certainty, and to use a human annotator on examples where the LLM has low certainty.

The paper discusses several different approaches to measuring uncertainty, and carry out experiments with each. Testing on three different tasks, the authors demonstrate that their approaches result in much stronger model performance after training on the produced data, when compared to a random allocation of examples between humans and LLMs.

Beyond implications for dataset construction, this paper is also interesting purely for the proposals on how to assess LLM uncertainty.

---

### Decision · Program_Chairs · 2023-10-07

**Decision:**

Accept-Main

**Comment:**

This paper proposes a framework for annotating data using both humans and LLMs. The core idea is to use the – cheaper – LLM to generate an annotation for examples where it has high certainty, and to use a human annotator on examples where the LLM has low certainty.

The paper discusses several different approaches to measuring uncertainty, and carry out experiments with each. Testing on three different tasks, the authors demonstrate that their approaches result in much stronger model performance after training on the produced data, when compared to a random allocation of examples between humans and LLMs.

Beyond implications for dataset construction, this paper is also interesting purely for the proposals on how to assess LLM uncertainty.